# Evaluation of a rapid lateral flow assay for the detection of taeniosis and cysticercosis at district hospital level in Tanzania: A prospective multicentre diagnostic accuracy study

Inge Van Damme[1,2☯], Chiara Trevisan[1,3☯], Mwemezi Kabululu[4], Dominik Stelzle[5], Charles E. Makasi[6,7], Veronika Schmidt-Urbaneja[5], Kabemba E. Mwape[8], Chishimba Mubanga[9], Gideon Zulu[8,10], Karen Shou Møller[11], Famke Jansen[12], Dries Reynders[13], John Noh[14], Sukwan Handali[14], Emmanuel Bottieau[15], Andrea S. Winkler[5,16,17], Pierre Dorny[12], Pascal Magnussen[18], Sarah Gabriël[1☯*], Bernard Ngowi[6,19☯]

1 Department of Translational Physiology, Infectiology and Public Health, Faculty of Veterinary Medicine, Ghent University, Merelbeke, Belgium, 2 Service Foodborne Pathogens, Sciensano, Brussels, Belgium, 3 Department of Public Health, Institute of Tropical Medicine, Antwerp, Belgium, 4 Tanzania Livestock Research Institute (TALIRI), Central Zone Office, Mpwapwa, Dodoma, Tanzania, 5 Department of Neurology, Center for Global Health, Technical University of Munich, Munich, Germany, 6 National Institute for Medical Research, Muhimbili Medical Research Centre, Dar es Salaam, Tanzania, 7 Kilimanjaro Christian Medical University College, Moshi, Tanzania, 8 Department of Clinical studies, School of Veterinary Medicine, University of Zambia, Lusaka, Zambia, 9 Department of Disease Control, School of Veterinary Medicine, University of Zambia, Lusaka, Zambia, 10 Ministry of Health, Government of the Republic of Zambia, Lusaka, Zambia, 11 Danish Veterinary and Food Administration, Glostrup, Denmark, 12 Department of Biomedical Sciences, Institute of Tropical Medicine, Antwerp, Belgium, 13 Department of Applied Mathematics, Computer Science and Statistics, Ghent University, Ghent, Belgium, 14 Parasitic Diseases Branch, Division of Parasitic Diseases, Center for Global Health, Centers for Disease Control and Prevention, Atlanta, Georgia, United States of America, 15 Department of Clinical Sciences, Institute of Tropical Medicine, Antwerp, Belgium, 16 Department of Community Medicine and Global Health, Institute of Health and Society, University of Oslo, Oslo, Norway, 17 Department of Global Health and Social Medicine, Harvard Medical School, Boston, Massachusetts, United States of America, 18 Department of Immunology and Microbiology, Faculty of Health and Medical Sciences, University of Copenhagen, Copenhagen, Denmark, 19 University of Dar es Salaam, Mbeya College of Health and Allied Sciences, Mbeya, Tanzania

☯ These authors contributed equally to this work.
* inge.vandamme@ugent.be

## Abstract

The TS POC test, *Taenia solium* point-of-care test, is a two-strip lateral flow assay using the recombinant antigen rES33 on the TS POC T test strip, and rT24H on the TS POC CC test strip, to detect antibodies against *T. solium* taeniosis and cysticercosis, respectively. The objective of this study was to assess the diagnostic performance of the TS POC test for the detection of *T. solium* taeniosis and cysticercosis in individuals attending district hospitals in Tanzania. In this prospective two-phase diagnostic accuracy study, we recruited participants aged 10 and above, excluding pregnant women and those with acute severe illness. Participants were consecutively recruited in three cohorts according to their signs/symptoms: compatible with neurocysticercosis (cohort 1), intestinal worm

**Data availability statement:** The data cannot be publicly shared due to ethical and privacy considerations but are accessible through the Data Access Committee at the Institute of Tropical Medicine, Antwerp. You can find more information and request access via https://www.itg.be/E/data-sharing-open-access, or via email at ITMresearchdataaccess@itg.be.

**Funding:** Funding for this research was provided by the European & Developing Countries Clinical Trials Partnership (grant number DRIA2014-308 to PD) and the German Federal Ministry of Education and Research (grant number 01KA1617 to ASW) as part of the research grant titled "Evaluation of an antibody detecting point-of-care test for the diagnosis of Taenia solium taeniosis and (neuro)cysticercosis in communities and primary care settings of highly endemic, resource-poor areas in Tanzania and Zambia, including training of and technology transfer to the Regional Reference Laboratory and health centers (SOLID)". The funders of this research had no role in study design, data collection and analysis, decision to publish, or preparation of the manuscript.

**Competing interests:** The authors have declared that no competing interests exist.

infections (cohort 2), and other signs/symptoms (cohort 3). Lacking a gold standard test for both infections, diagnostic accuracy was evaluated using results of two coprological and two serological tests for taeniosis, and three serological tests for cysticercosis, in a Bayesian Latent Class Model approach. The TS POC test was conducted on 601 participants in cohort 1, 1661 participants in cohort 2, and 662 participants in cohort 3. Most individuals tested negative on both TS POC test strips, with proportions of 83% (n = 496), 97% (n = 1613) and 97% (n = 641) in cohorts 1, 2 and 3, respectively. Complete case data were available for 120, 114, and 53 participants for taeniosis, and 126, 122, and 55 participants for cysticercosis. Sensitivity values for the TS POC T test strip were 50.2% [95% credible interval 4.9 - 96.4], 40.8% [2.2 - 95.2], and 40.4% [2.3 – 95.0], while specificity values were 98.6% [97.1 - 99.6], 99.3% [98.7 - 99.7] and 99.4% [98.5 - 99.9], respectively. For the TS POC CC test strip, the sensitivity was 77.5% [37.8 - 99.2], 24.9% [95% CI 6.4 - 52.7] and 44.2% [6.6 - 91.5], and the specificity 92.3% [86.5 - 98.8], 99.1% [97.8 - 100], and 98.1% [96.1 - 99.7] across the respective cohorts. Although the TS POC test has a low sensitivity, it demonstrates a high specificity, which may have clinical utility to guide treatment and diagnostic decisions, or in epidemiological studies. An important strength of this study lies in its assessment of the TS POC test under real-world conditions, revealing divergent estimates across distinct cohorts. The study underscores the suboptimal performance of existing tests under field conditions, emphasizing the need to enhance and validate these tests for better performance in practical real-world settings.

**Registration number:** PACTR201712002788898.

## Author summary

*Taenia solium* poses significant public health concerns globally and is a leading cause of acquired epilepsy in *T. solium* endemic areas. The parasite causes two distinct infections in humans: taeniosis, an intestinal infection, and cysticercosis, a tissue infection. The disease is particularly prevalent in low-resource settings, contributing to substantial morbidity and economic burdens. Recently, a test was specifically developed as an affordable and rapid diagnostic tool, tailored for deployment in resource-constrained regions. The TS POC test is composed of two test strips, one to detect taeniosis and one for cysticercosis. This study provides a critical assessment of the diagnostic efficacy of the TS POC test in Tanzanian district hospital settings. By evaluating the test's performance across diverse cohorts and real-world conditions, the research sheds light on the limitations of existing diagnostic modalities and underscores the imperative for improved testing strategies. The findings offer valuable insights for public health practitioners and policymakers striving to enhance diagnostic capabilities and ultimately mitigate the burden of *T. solium* infections in endemic regions.

## 1 Introduction

*Taenia solium* is an important zoonotic parasite, affecting humans and pigs. In humans, infections can have different presentations. The adult tapeworm resides in the intestines of humans (taeniosis), while its larvae can lead to a systemic infection (cysticercosis, CC).

Humans become infected with the tapeworm by consuming undercooked or raw pork that contains viable cysticerci. When eggs of the tapeworm are ingested, larvae can encyst in various tissues of the human body, including the central nervous system, causing neurocysticercosis (NCC). The impact of *T. solium* on public health, economics, and social welfare is significant, particularly in resource-limited settings. In Tanzania, the prevalence of *T. solium* taeniosis and cysticercosis is high, with prevalence rates of up to 5% in stool and 45% having antibodies based on serology, respectively [1,2], resulting in significant public health and economic consequences [3].

Diagnosis of *T. solium* taeniosis and cysticercosis in humans is currently performed through laboratory-based methods targeting antigens and antibodies, such as enzyme-linked immunosorbent assay (ELISA) and immunoelectrotransfer blot (EITB) assay [4,5], and also basic coprological methods for taeniosis [6]. If there is a suspicion of NCC, either on clinical grounds and/or positive cysticercosis serology, neuroimaging such as computed tomography is the diagnostic tool of choice. Also, several molecular methods are available for the diagnosis of taeniosis and the confirmation of cysticercosis lesions, which are mostly PCR-based [4,7]. Most of these methods are time-consuming, expensive and/or not readily available in rural areas, leading to a significant challenge in the management of these infections [8]. Rapid diagnostic tests (RDTs) have revolutionized the field of diagnostics by providing fast and easy-to-use tools for the detection of diseases in resource-limited settings. The implementation of RDTs has the potential to significantly reduce the global burden of disease, as evidenced by studies estimating the impact on acute lower respiratory infections [9], human immunodeficiency virus [10], and tuberculosis [11]. However, the accuracy of RDTs is often questioned, as field validation is often not performed [12].

For taeniosis, a rapid test that can be performed at the point of care (POC) would facilitate timely and targeted treatment, reducing disease transmission. Furthermore, an easy and rapid test for the detection of cysticercosis may be useful in patient care, disease monitoring programs and epidemiological studies. As such, an easy-to-use POC test (called TS POC test) has been developed for *T. solium* by the Centers for Disease Control and Prevention (CDC) and the Technical University of Munich (TUM). The test showed a promising sensitivity and specificity to detect *T. solium* taeniosis and NCC during its preliminary evaluation under laboratory conditions. To evaluate its real-world performance in resource-poor, highly endemic areas in sub-Saharan Africa, studies were designed to evaluate the performance characteristics of the TS POC test in two different settings: in rural communities in Zambia [13], and in district hospitals in Tanzania [14]. The evaluation of the TS POC test for the detection of *T. solium* taeniosis and cysticercosis at community level has been reported elsewhere [15,16]. Also the evaluation of the TS POC test for the neuroimaging-based diagnosis of NCC in both settings were reported elsewhere [17,18], and are thus out of the scope of this paper. The focus of this paper is the evaluation of the TS POC test for the detection of infection with the adult tapeworm (taeniosis) and/or the larval stage (cysticercosis) of *T. solium* in individuals attending district hospitals (Tier 2 level) in Tanzania. The primary endpoints of this paper are the sensitivity and specificity of the TS POC test for the detection of both infections, evaluated in three different clinical cohorts.

## 2 Materials and methods

### 2.1 Ethics statement

The study adhered to the principles outlined in the Declaration of Helsinki and obtained ethical approval from the National Ethics Health Research Committee (NatREC) of Tanzania (NIMR/HQ/R.8a/Vol.IX/2597), the Institute of Tropical Medicine (IRB/AB/ac/112 Ref

1177/17) through the ethics committee of the University of Antwerp (EC UZA 17/31/352), and the Technical University of Munich via their Ethics Committee at the Klinikum rechts der Isar, Munich (299/18S). Written informed consent (assent for minors, with written consent of parent/guardian) was obtained from all participants.

## 2.2 Trial registration

The trial was registered at the Pan African Clinical Trials Registry with identifier PACTR201712002788898. A comprehensive description of the trial rationale, design and methodology can be found in Trevisan et al. [14]. Only the methodological details relevant for this manuscript are summarized in this paper, i.e., only focusing on *T. solium* taeniosis and cysticercosis testing. The evaluation of the TS POC CC test strip for NCC diagnosis is described elsewhere [17,18].

## 2.3 Study design

The study was designed as a prospective, two-phase, multicentre diagnostic accuracy study. First, all participants were tested using the TS POC test (phase 1), after which all participants testing positive and a subset of the participants testing negative were requested to provide a blood and stool sample for further testing (phase 2). Since there is no gold standard as comparison method for either of the infections, the performance of the TS POC test was assessed using a combination of tests. The details of these tests can be found in section 2.4, sample collection and testing. The TS POC test is a two-strip lateral flow assay prototype. The primary aim of this paper is to determine the sensitivity and specificity of the TS POC T test strip and TS POC CC test strip for the detection of *T. solium* taeniosis and cysticercosis, respectively, in individuals attending district hospitals in Tanzania. As sensitivity and specificity are known to differ between study populations, the evaluation was done in three different cohorts: 1) individuals with specific neurological signs and symptoms compatible with NCC (epilepsy and/or severe progressive headache) (from now on referred to as « cohort 1 »); 2) individuals with complaints compatible with intestinal worm infections such as abdominal pain and loss of appetite (cohort 2); and 3) individuals with other signs/symptoms, such as coughing, limb pain, hypertension, and diabetes patients (cohort 3).

## 2.4 Participants

Participants were enrolled between December 2017 and February 2020 in three district hospitals in southern Tanzania: Mbeya (rural) District (Ifisi) and Rungwe District (Tukuyu), both located in Mbeya Region, and Mbozi District (Vwawa), located in Songwe Region. To be included, individuals aged 10 years or above had to be willing and able to provide written informed consent, living in the study area for the past three months and planning to stay in the same area throughout the study period. Pregnant women and individuals with acute severe illness that needed in-patient care were excluded. Different additional inclusion criteria were applied to define eligibility for the different cohorts, as described below.

Individuals with neurological signs/symptoms compatible with NCC (cohort 1) were consecutively recruited from Outpatient Departments (OPD) and the Mental Health Clinics (MHC). A questionnaire consisting of nine questions was used to assess if individuals fulfilled the criteria for epileptic seizures and/or severe progressive headache (S1 Appendix; more details can be found in Stelzle et al. [17,19]). For cohort 2, individuals who presented at the OPD with complaints compatible with intestinal worm infection were included. They needed to have at least one of the following signs/symptoms: abdominal pain/discomfort, having seen worm parts in the stool, express having a worm in the stomach, nausea, diarrhoea

and loss of appetite. All other people presenting at the OPD, without signs/symptoms compatible with cohort 1 or cohort 2, were potential candidates to be included in cohort 3. Participants with epilepsy and/or severe progressive headache (cohort 1) or signs/symptoms compatible with intestinal worm infections (cohort 2) were consecutively recruited. For cohort 3, every 10th individual was approached for enrolment in the study.

Eligible participants were informed about the study and were invited to provide informed consent. Written informed consent (assent for minors, with written consent of parent/guardian) was obtained from all participants. Following consent, demographic and clinical data were collected from participants, including self-reported age, and gender.

## 2.5  TS POC test

The TS POC is an antibody-detecting lateral flow assay, using two test strips, each with a previously characterised recombinant protein, rES33 for the TS POC T test strip and rT24H for the TS POC CC test strip [20,21]. During initial assessments conducted within laboratory settings using known positive and negative control sera, the TS POC test demonstrated encouraging results. The TS POC T test strip exhibited a sensitivity of 82% and a specificity of 99% in detecting taeniosis. In the case of NCC, the TS POC CC test strip had a sensitivity of 88%, and 93% for infection with multiple cysticerci, with a corresponding specificity of 99%. The TS POC cassette prototype was assembled at CDC, Atlanta.

Details about the TS POC test and testing procedures have been described elsewhere [13,15,16,22]. In short, 20 µL blood was collected from a fingertip and placed in a sample port of one of the test strips, and the same procedure was repeated for the other test strip. After applying chase buffer, the TS POC test was read after 20 minutes. A TS POC test strip was considered positive when the control line and test line were positive (visible as a red line). The result was invalid when the control line was negative. The result was negative when the control line was positive and test line negative. The results were read by two different readers. When there was disagreement between the readers, the result of a third reader was decisive.

## 2.6  Sample collection and testing

Participants testing positive for one or both TS POC test strips, and every 10th participant testing negative on both strips, were further sampled and tested. Since there is no gold standard test available for taeniosis or cysticercosis, multiple imperfect tests were used. For taeniosis, the following three tests were used: 1) copro Ag ELISA according to Allan et al. [23] and modified by Mwape et al. [24] to detect *Taenia* antigens in stool using a predefined cut-off value of 0.972; 2) copro mPCR according to Yamasaki et al. [7] to detect *Taenia* DNA in stool, which was considered positive only when a 720 bp *T. solium* band was visible; and 3) rES33-EITB to detect antibodies in serum [25], which was positive when the rES33 band was visible. For cysticercosis, two serum-based tests were used: 1) serum Ag ELISA to detect antigens with the cut-off value determined by comparing the optical density to that of a set of negative serum samples according to Dorny *et al.* [26], and 2) rT24H-EITB to detect antibodies [20], which was considered positive when the rT24H band was visible. Although also the LLGP-EITB was initially planned to be included in the study [14], this in-house test could not be used due to doubts about the validity of the test results. The test did not satisfy internal quality control standards during laboratory analyses [16], so it was decided to exclude the results to ensure the reliability of the results presented in this study. Further details regarding the tests are available in the study protocols [13,14]. No adverse events were recorded during the collection of blood samples for the serological tests.

Samples were shipped from Tanzania to Belgium for analysis. The PCR assays, copro Ag ELISA and immunoblots were performed at the Institute of Tropical Medicine (Antwerp, Belgium). Serum Ag ELISA was performed at Ghent University (Merelbeke, Belgium). Samples arriving at the laboratories were only labelled with a pseudonymized code, so laboratory personnel performing the tests were blinded to the TS POC result and the participant's cohort.

## 2.7 Statistical analysis

**2.7.1 Bayesian analysis.** The primary objective of this paper was to determine the sensitivity and specificity of: 1) the TS POC T test strip for the detection of taeniosis, and 2) the TS POC CC test strip for the detection of cysticercosis. Due to the lack of a gold standard test for infection with adult worm and larval stages of *T. solium*, diagnostic accuracy measures were estimated using a Bayesian Latent Class Model (BLCM)-like approach according to Berkvens et al. [27] using Open BUGS software version 3.2.3 (www.openbugs.net). The analysis allowed the sensitivities and specificities to differ conditional on the other test results. The R code used for the analyses is provided in S2 Appendix. The analyses were performed for each of the three cohorts separately. Directed acyclic graphs are included in S3 Appendix to visualise the relationship between the target conditions and the different diagnostic tests.

This study was reported following the Standards for Reporting of Diagnostic Accuracy studies that use BLCMs (STARD-BLCM) [28].

Sensitivity and specificity of the TS POC test were considered co-primary endpoints. The positive/negative predictive values, the prevalence, and the accuracy measures of the tests were also estimated using the models and were considered exploratory endpoints.

**2.7.1.1 Handling missing and inconclusive results:** The test strips of the TS POC test were read by two readers, and a third reader in case the first two readers disagreed. There were no inconclusive TS POC results found. For cysticercosis, participants with one or more missing blood results were excluded from the analysis, and for taeniosis, participants with a missing result for a blood test and/or stool test were excluded, i.e., a complete case analysis was performed. To avoid partial verification bias due to the two-phase sampling, the multinomial probabilities in the models were adapted according to the observed sampling frequencies. The weighting was determined by the proportion of complete cases per test strip result.

**2.7.1.2 Priors:** For the probabilistic constraints in the Bayesian analyses, initially the same priors were used as described previously, for the evaluation of the TS POC test accuracy at community level Click or tap here to enter text [15,16]. However, the priors were revised since diagnostic accuracy measures differ according to the target population, and the data did not support the priors in certain models, as demonstrated by high Bayesian P values (see S3 Appendix for all model outcomes). Therefore, new priors were defined based on the knowledge that was obtained about the test performance during the field studies at community level [15,16]. Given the well-documented bias of sensitivity and specificity estimates from laboratory-based case-control studies, and the lack of field validation data, this approach was deemed the most appropriate. The outputs of the least restrictive models of the community-based studies were used as a basis to determine the new sets of priors and were updated after a more thorough literature search whenever necessary. The priors that were used during the first (initial priors) and second (new priors) round of analyses, including the rationale, are given in S3 Appendix.

**2.7.1.3 Models:** Only the outputs of the final models are reported in the main article for clarity, but the output of all models that were performed can be found in S3 Appendix for completeness. The output of the new sets of priors was used as the final model for most analyses. Only for the evaluation of cysticercosis tests in cohort 2, the results from the model using the least restrictive original priors were used because this model fitted considerably better

compared to the model using the new priors (Bayesian p value of 0.635 and 0.790, respectively; see S3 Appendix).

Due to the low number of positive cases for taeniosis, all analyses for taeniosis were repeated for the three cohorts combined. When combining the three cohorts, one overall prevalence (for the mixture of the three cohorts) was estimated, and the conditional sensitivities and specificities were assumed to be equal. The overall models (combining the three cohorts) are included in S3 Appendix, but they are not reported in the main manuscript due to the high Bayesian p values, indicating a bad fit.

For the evaluation of the TS POC T strip, the TS POC CC test strip result was included in all models to account for the relatively large number of TS POC CC positive participants within TS POC T negative complete cases. Since rES33-EITB detects exposure (antibodies) whereas copro mPCR and copro Ag ELISA detect infection (parasitic DNA and proteins, respectively) and the TS POC T test was developed to identify active infections by detecting higher levels of antibodies, the analyses were also repeated without rES33-EITB to estimate the accuracy for detecting active infection. The estimates from models with and without rES33-EITB were very similar (see S3 Appendix), so only the models including rES33-EITB are reported as final model for taeniosis.

### 2.7.2 Agreement between tests

Cohen's kappa statistics, and positive/negative agreements were calculated to explore the agreement between different tests [29]. The observed frequencies of the complete cases were inversely weighted according to their sampling frequencies to calculate the measures of agreement.

### 2.7.3 Sample size and descriptive statistics

The sample sizes were calculated to obtain a desired precision of 10% around the sensitivity and specificity of the TS POC test, resulting in 600 individuals in cohort 1, and 2000 individuals, distributed over cohort 2 and 3 [14]. Demographic characteristics of participants were reported descriptively, using R version 4.2.3 [30].

## 3 Results

### 3.1 Study population

In total, 3055 participants were recruited in three district hospitals in Tanzania. Participants were recruited for neurological signs/symptoms compatible with NCC (cohort 1, n = 742); complaints compatible with intestinal worm infection (cohort 2; n = 1661); and other symptoms (cohort 3; n = 663). The flows for each of the three cohorts are visualised in Figs 1–3, respectively. The TS POC test was performed in 601 participants of cohort 1, 1661 participants of cohort 2, and 662 participants of cohort 3. The baseline characteristics of these participants in each of the three cohorts are given in Table 1. The median age varied between 33 and 40 years and the proportion of women varied between 52% and 69%. The characteristics of patients who were selected for sampling, and of the participants in the final analysis set, are provided in S1 Appendix.

In cohort 1, 422 participants were recruited from the mental health clinic, and 170 from the outpatient department (S1 Appendix). Based on the initial screening questionnaire administered by local nurses, participants were recruited for both severe progressive headache and epilepsy (n = 307), only epilepsy (n = 169) or only severe progressive headache (n = 125). An overview of symptoms based on the screening questionnaires is given in S1 Appendix. More details regarding the characteristics and neurological signs/symptoms of participants in cohort 1 can be found in Stelzle *et al.* [19].

**Table 1. Baseline characteristics of participants tested using the TS POC test, in each of the three cohorts, recruited in three district hospitals in Tanzania.**

|  | Cohort 1 | Cohort 2 | Cohort 3 |
|---|---|---|---|
|  | (n = 601) | (n = 1661) | (n = 662) |
| Age (in years) |  |  |  |
| Mean | 36 | 38 | 43 |
| Median [interquartile range] | 33 [23] | 33 [32] | 40 [28] |
| Gender |  |  |  |
| Female (%) | 313 (52%) | 1146 (69%) | 445 (67%) |
| Male (%) | 288 (48%) | 515 (31%) | 217 (33%) |

Cohort 1: people with specific neurological signs/symptoms (epilepsy and/or severe progressive headache); cohort 2: people with complaints compatible with intestinal worm infections; cohort 3: people with any other symptom(s).

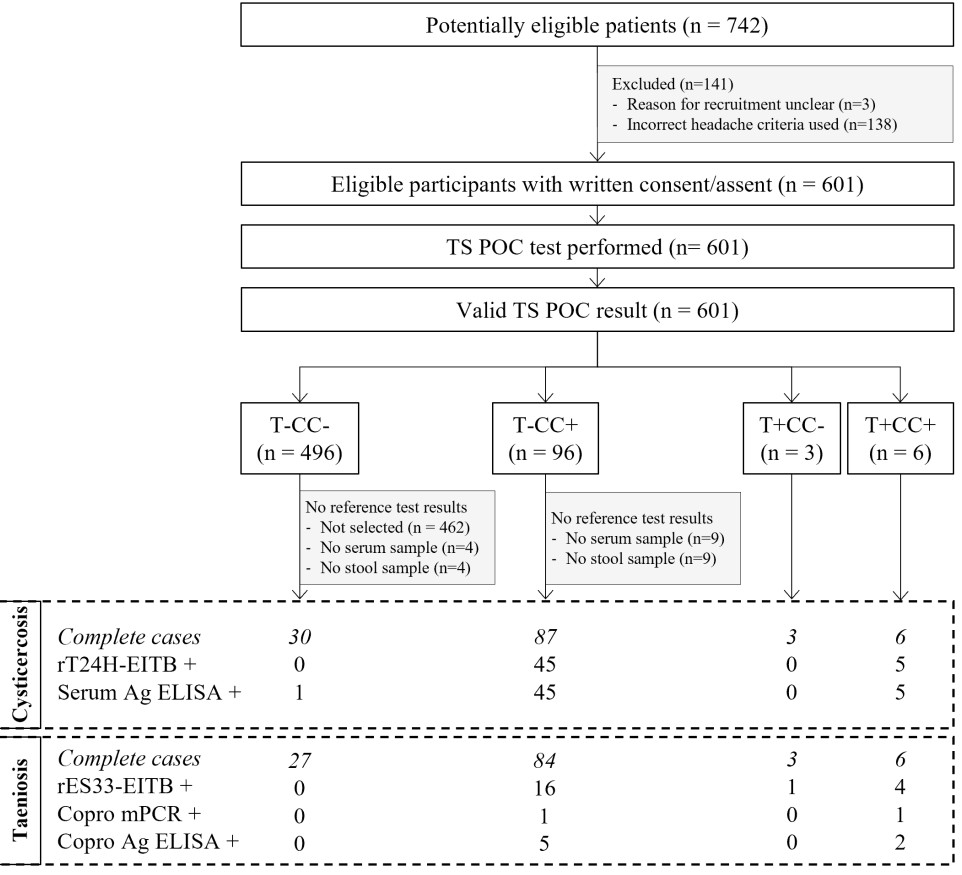

**Fig 1. Flow diagram of participants with specific neurological signs/symptoms compatible with NCC (cohort 1).** TS POC: *T. solium* point-of-care test; rT24H-EITB: recombinant T24H enzyme-linked immunoelectrotransfer blot, serum Ag ELISA: enzyme-linked immunosorbent assay detecting *Taenia* antigens in serum; rES33-EITB: recombinant ES33 enzyme-linked immunoelectrotransfer blot, copro mPCR: multiplex polymerase chain reaction in stool, copro Ag ELISA: enzyme-linked immunosorbent assay detecting *Taenia* antigens in stool. The numbers after each of the taeniosis and cysticercosis test results refer to the number of positive samples.

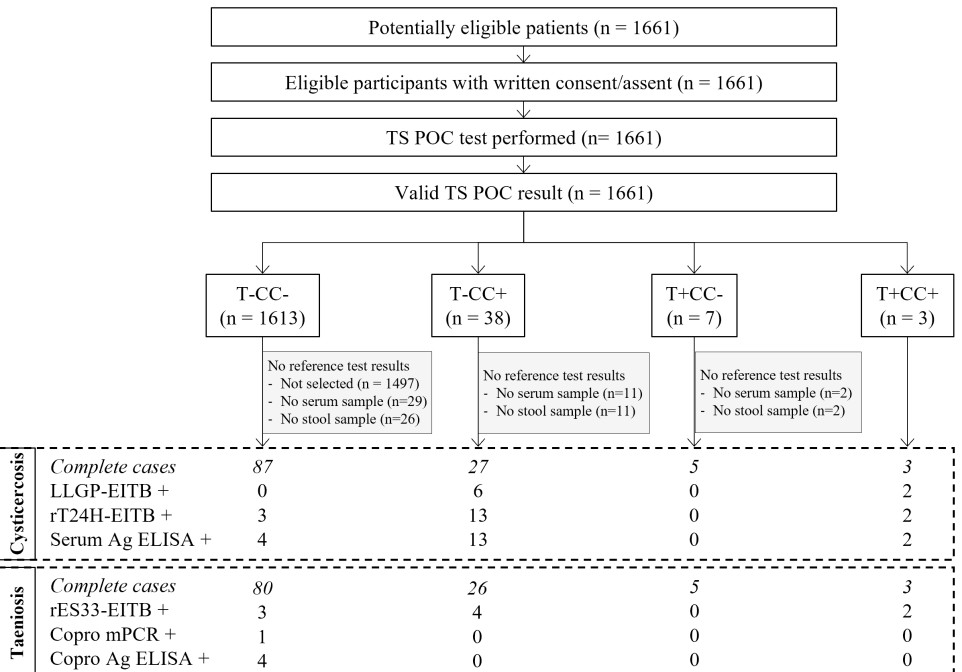

**Fig 2. Flow diagram of participants with complaints compatible with intestinal worm infections (cohort 2).** TS POC: *T. solium* point-of-care test; rT24H-EITB: recombinant T24H enzyme-linked immunoelectrotransfer blot, serum Ag ELISA: enzyme-linked immunosorbent assay detecting *Taenia* antigens in serum; rES33-EITB: recombinant ES33 enzyme-linked immunoelectrotransfer blot, copro mPCR: multiplex polymerase chain reaction in stool, copro Ag ELISA: enzyme-linked immunosorbent assay detecting *Taenia* antigens in stool. The numbers after each of the taeniosis and cysticercosis test results refer to the number of positive samples.

In cohort 2, participants were included with gastro-intestinal symptoms. Out of the participants with complete data for the screening questionnaire (n = 1537), most indicated to have three (n = 595) or two (n = 568) symptoms, and fewer had one (n = 252), four (n = 117) or five (n = 5) symptoms. Most participants reported abdominal pain/discomfort (n = 1438), followed by loss of appetite (n = 827), nausea (n = 796), diarrhoea (n = 403), express having a worm in the stomach (n = 110) and having seen worm parts in their stool (n = 92).

## 3.2 TS POC test results

Within the three cohorts, most participants tested negative using both TS POC test strips (n = 496, 83%; n=1613, 97%; n=641, 97% in cohort 1, 2 and 3, respectively). All participants who were positive for at least one test strip and a subset of participants negative using both test strips were selected to give a blood and stool sample for further testing. Overall, 287 complete cases, *i.e.,* results available for all tests, were obtained for the evaluation of taeniosis, and 303 complete cases were obtained for cysticercosis. Details regarding the TS POC result combinations within each of the cohorts can be found in the respective flow diagrams (Figs 1-3).

## 3.3 Taeniosis tests

Most participants provided a stool sample immediately after recruitment, with a median duration of two days between the TS POC test and processing of the stool sample in the hospital lab (based on 220 participants for whom both dates were known). Nevertheless, 27 participants provided a stool sample more than one month after recruitment.

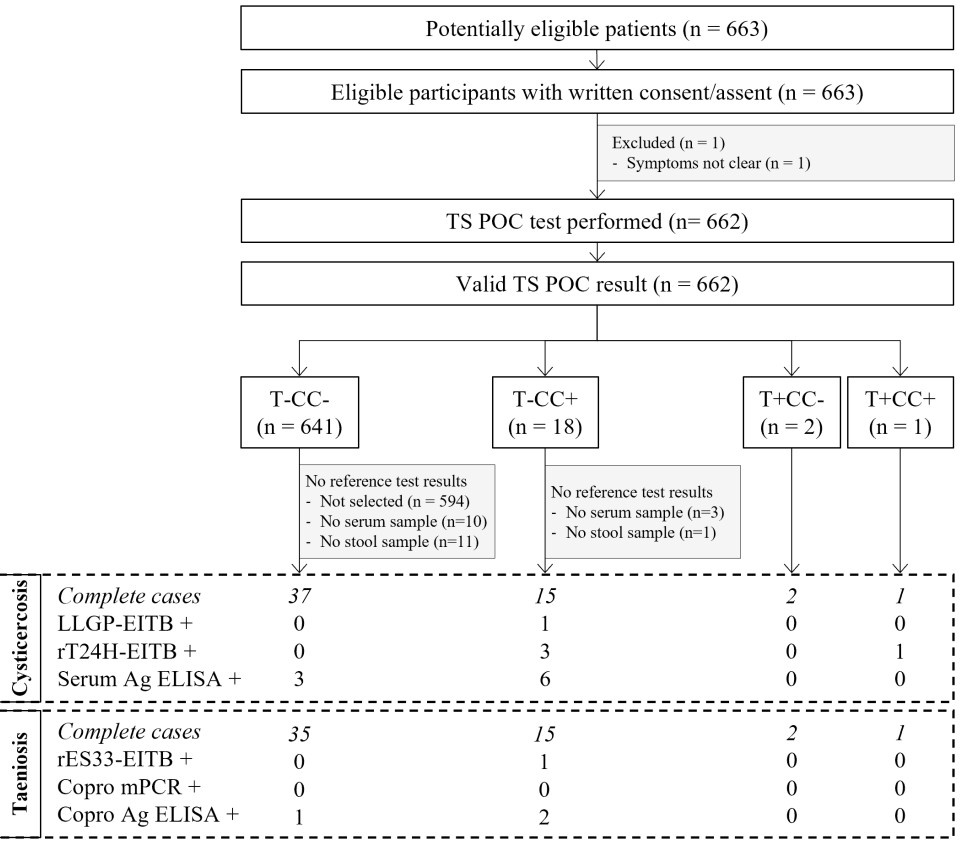

**Fig 3. Flow diagram of participants attending district hospitals with symptoms other than neurological and gastro-intestinal (cohort 3).** TS POC: *T. solium* point-of-care test; rT24H-EITB: recombinant T24H enzyme-linked immunoelectrotransfer blot, serum Ag ELISA: enzyme-linked immunosorbent assay detecting *Taenia* antigens in serum; rES33-EITB: recombinant ES33 enzyme-linked immunoelectrotransfer blot, copro mPCR: multiplex polymerase chain reaction in stool, copro Ag ELISA: enzyme-linked immunosorbent assay detecting *Taenia* antigens in stool. The numbers after each of the taeniosis and cysticercosis test results refer to the number of positive samples.

**3.3.1 Test results.** Table 2 shows the results of the 287 complete cases for the evaluation of *T. solium* taeniosis, per cohort. Within TS POC T negative participants, the majority (231/267, 87%) tested negative using all other taeniosis tests. Also, within TS POC T positive complete cases, most participants (12/20, 60%) tested negative using all other tests.

**3.3.2 Diagnostic performance measures.** Based on the output of the final model for taeniosis, the TS POC T test strip had an estimated sensitivity of 50.2% [95% CI 4.9 - 96.4] in cohort 1, 40.8% [2.2 - 95.2] in cohort 2, and 40.4% [2.3 - 95] in cohort 3. The specificity was 98.6% [97.1 - 99.6], 99.3% [98.7 - 99.7] and 99.4% [98.5 - 99.9], respectively. The sensitivity and specificity of the TS POC T test strip, rES33-EITB, copro Ag ELISA and copro mPCR are visualised in Fig 4. The estimated sensitivities of all tests were low, ranging from 40.8% to 56.5%, and all had wide credible intervals. The prevalence of *T. solium* taeniosis in the cohorts was estimated at 1.1% [0.1 - 3.5] (cohort 1), 0.3% [0 - 1.4] (cohort 2), and 0.5% [0 - 2.4] (cohort 3). The positive predictive values of the TS POC T test strip were 25% [0.9 - 71.6], 10% [0.2 - 41.2], and 20.1% [0.4 - 73.1], and the negative predictive values were 99.4% [97.2 - 100], 99.8% [98.8 - 100], 99.6% [97.9 - 100], and 99.8% [98.9 - 100] for the three cohorts, respectively (S3 Appendix).

**Table 2. Cross tabulation of the tests used to detect taeniosis within the three cohorts (complete cases only, n = 287).**

| TS POC T test strip | rES33-EITB | Copro mPCR | Copro Ag ELISA | Cohort 1 | | Cohort 2 | | Cohort 3 | | Total |
|---|---|---|---|---|---|---|---|---|---|---|
| | | | | TS POC CC- | TS POC CC+ | TS POC CC- | TS POC CC+ | TS POC CC- | TS POC CC+ | |
| negative | negative | negative | negative | 27 | 63 | 73 | 22 | 34 | 12 | 231 |
| negative | negative | negative | positive | 0 | 4 | 3 | 0 | 1 | 2 | 10 |
| negative | negative | positive | positive | 0 | 1 | 1 | 0 | 0 | 0 | 2 |
| negative | positive | negative | negative | 0 | 16 | 3 | 4 | 0 | 1 | 24 |
| positive | negative | negative | negative | 2 | 1 | 5 | 1 | 2 | 1 | 12 |
| positive | negative | negative | positive | 0 | 1 | 0 | 0 | 0 | 0 | 1 |
| positive | positive | negative | negative | 1 | 3 | 0 | 2 | 0 | 0 | 6 |
| positive | positive | positive | positive | 0 | 1 | 0 | 0 | 0 | 0 | 1 |

TS POC T: *T. solium* point-of-care test strip for taeniosis; rES33-EITB: recombinant ES33 enzyme-linked immunoelectrotransfer blot, copro mPCR: multiplex polymerase chain reaction in stool, copro Ag ELISA: enzyme-linked immunosorbent assay detecting *Taenia* antigens in stool. The numbers within each cohort represent the number of participants for each TS POC T and TS POC CC test strip result combination. A two-phase design was used, selecting only a subset of the TS POC negative population for further testing, so particularly participants that were negative using both TS POC test strips are underrepresented. Combinations of test results that did not occur in the dataset are not included in the table.

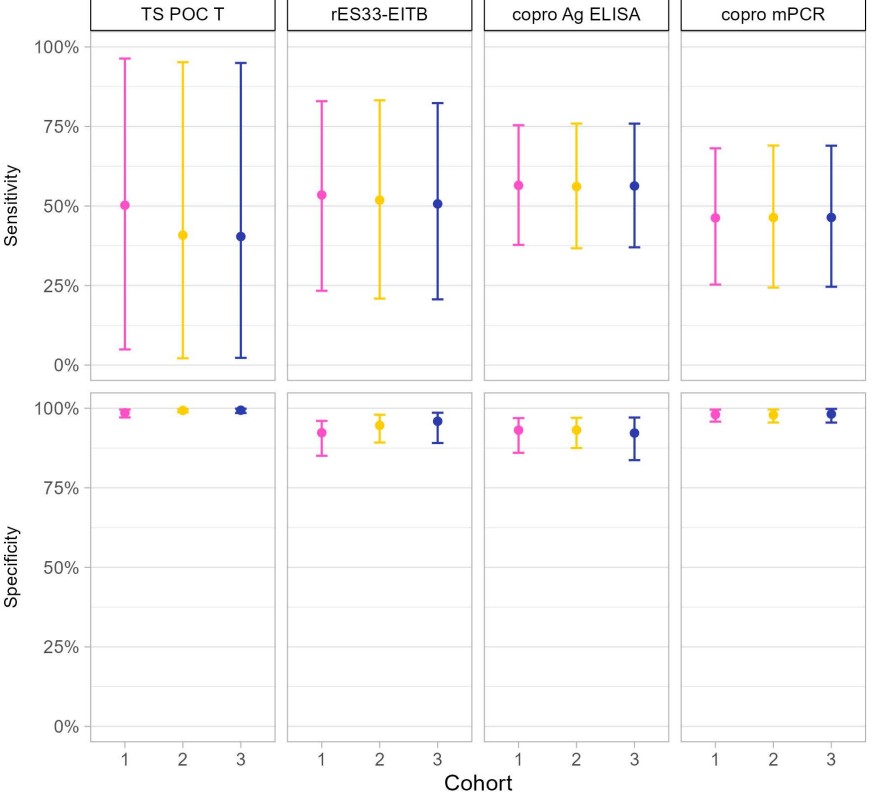

**Fig 4. Sensitivity and specificity of the TS POC T test strip, rES33-EITB, copro Ag ELISA and copro mPCR to detect taeniosis in three cohorts of participants recruited at district hospitals in Tanzania.** Results of the same cohort are indicated with the same colour. The error bars indicate the 95% credible intervals around the mean estimates. Cohort 1: participants with specific neurological signs and symptoms compatible with NCC (epilepsy and/ or severe progressive headache); cohort 2: participants with complaints compatible with intestinal worm infections; cohort 3: participants with other symptom(s). TS POC T: *T. solium* point-of-care test strip for taeniosis; rES33-EITB: recombinant ES33 enzyme-linked immunoelectrotransfer blot, copro Ag ELISA: enzyme-linked immunosorbent assay detecting *Taenia* antigens in stool, copro mPCR: multiplex polymerase chain reaction in stool.

### 3.3.3 Agreement between taeniosis tests

The point estimates for agreement and positive agreement were low among all tests for taeniosis (Table 3). Due to the low number of test positive samples, the estimates for agreement should be interpreted with caution. There was no agreement between the TS POC T test and any of the other tests within cohort 2 and cohort 3. Also, the agreement between rES33-EITB and the stool-based tests was low, with positive agreements from 0 to 8%. Only within cohort 1, the agreement between the TS POC T test strip and rES33-EITB was fair, with 31% positive agreement. All three copro mPCR positive cases within this cohort were also positive using copro Ag ELISA.

## 3.4 Cysticercosis

**3.4.1 Test results.** The results of cysticercosis tests for the complete cases within each of the three cohorts is shown in Table 4. Within TS POC CC negative complete cases, only few were positive using either serum Ag ELISA or rT24H-EITB, and none of the POC CC negative participants tested positive using both other tests simultaneously. Within TS POC CC positive complete cases, most participants tested positive using both tests or negative using both tests (Table 4).

**Table 3. Agreement between the different tests for taenioisis and cysticercosis, for the three different cohorts of participants.**

|  | Cohort 1 | Cohort 2 | Cohort 3 |
|---|---|---|---|
| **Taeniosis tests** |  |  |  |
| TS POC T - rES33-EITB | 0.29 \| 0.31 \| 0.98 | 0.04 \| 0.05 \| 0.98 | 0.00 \| 0.00 \| 1.00 |
| TS POC T – copro mPCR | 0.17 \| 0.18 \| 0.99 | -0.01 \| 0.00 \| 0.99 | 0.00 \| 0.00 \| 1.00 |
| TS POC T – copro Ag ELISA | 0.23 \| 0.24 \| 0.99 | -0.01 \| 0.00 \| 0.97 | -0.01 \| 0.00 \| 0.98 |
| rES33-EITB – copro mPCR | 0.07 \| 0.08 \| 0.98 | -0.02 \| 0.00 \| 0.97 | 0.00 \| 0.00 \| 1.00 |
| rES33-EITB – copro Ag ELISA | 0.05 \| 0.06 \| 0.98 | -0.04 \| 0.00 \| 0.95 | 0.00 \| 0.00 \| 0.98 |
| Copro mPCR – copro Ag ELISA | 0.43 \| 0.43 \| 1.00 | 0.39 \| 0.40 \| 0.98 | 0.00 \| 0.00 \| 0.98 |
| **Cysticercosis tests** |  |  |  |
| TS POC CC - rT24H-EITB | 0.66 \| 0.70 \| 0.95 | 0.33 \| 0.35 \| 0.98 | 0.38 \| 0.39 \| 0.99 |
| TS POC CC – serum Ag ELISA | 0.57 \| 0.63 \| 0.94 | 0.27 \| 0.30 \| 0.97 | 0.15 \| 0.18 \| 0.95 |
| rT24H-EITB – serum Ag ELISA | 0.74 \| 0.76 \| 0.97 | 0.18 \| 0.22 \| 0.96 | 0.10 \| 0.11 \| 0.96 |

The numbers indicate Cohen's kappa, positive and negative agreement, respectively. Point estimates were calculated based on complete cases, which were inversely weighted according to their observed sampling frequencies (according to the TS POC C and TS POC T result combination) to account for the study design.

**Table 4. Cross tabulation of the tests used to detect cysticercosis within the three cohorts of participants (complete cases only; n = 303).**

| TS POC CC test strip | rT24H-EITB | Serum Ag ELISA | Cohort 1 | | Cohort 2 | | Cohort 3 | | Total |
|---|---|---|---|---|---|---|---|---|---|
|  |  |  | POC T- | POC T+ | POC T- | POC T+ | POC T- | POC T+ |  |
| negative | negative | negative | 29 | 3 | 80 | 5 | 34 | 2 | 153 |
| negative | negative | positive | 1 | 0 | 4 | 0 | 3 | 0 | 8 |
| negative | positive | negative | 0 | 0 | 3 | 0 | 0 | 0 | 3 |
| positive | negative | negative | 36 | 1 | 13 | 1 | 9 | 0 | 60 |
| positive | negative | positive | 6 | 0 | 1 | 0 | 3 | 0 | 10 |
| positive | positive | negative | 6 | 0 | 1 | 0 | 0 | 1 | 8 |
| positive | positive | positive | 39 | 5 | 12 | 2 | 3 | 0 | 61 |

TS POC CC: *T. solium* point-of-care test strip for cysticercosis; rT24H-EITB: recombinant T24H enzyme-linked immunoelectrotransfer blot, serum Ag ELISA: enzyme-linked immunosorbent assay detecting *Taenia* antigens in serum. The numbers within each cohort represent the number of participants for each result combination. A two-phase design was used, selecting only a subset of the TS POC negative population for further testing, so participants that were negative on both TS POC test strips are underrepresented.

**3.4.2 Diagnostic performance of cysticercosis tests.** The complete cases in Table 4 were used to estimate the performance characteristics of the TS POC CC test strip using a Bayesian model, for each of the three cohorts separately. The sensitivity and specificity of the final models are visualised in Fig 5. The sensitivity of the TS POC CC test strip was 77.5% [37.8 - 99.2] in cohort 1, 24.9% [6.4 - 52.7] in cohort 2, and 44.2% [2.3 - 95] in cohort 3. The specificity was 92.3% [86.5 - 98.8], 99.1% [97.8 - 100], and 98.1% [96.1 - 99.7], respectively. The point estimates of the sensitivity of all tests were higher in cohort 1 than in the other two cohorts. The prevalence of cysticercosis was estimated at 15.3% [8.1 - 29.4] (cohort 1), 9.4% [5.1 - 23.9] (cohort 2), and 5.1% [0.8 - 17.1] (cohort 3). Positive predictive values of the TS POC CC test strip were 63.3% [44.1 - 94.5], 71.6% [35.4 - 98.8], and 46.6% [13.5 - 91.2], and negative predictive values 95.1 [79.4 - 99.9], 92.4% [77.4 - 97.3], and 96.4% [84.2 - 99.9], respectively (S3 Appendix).

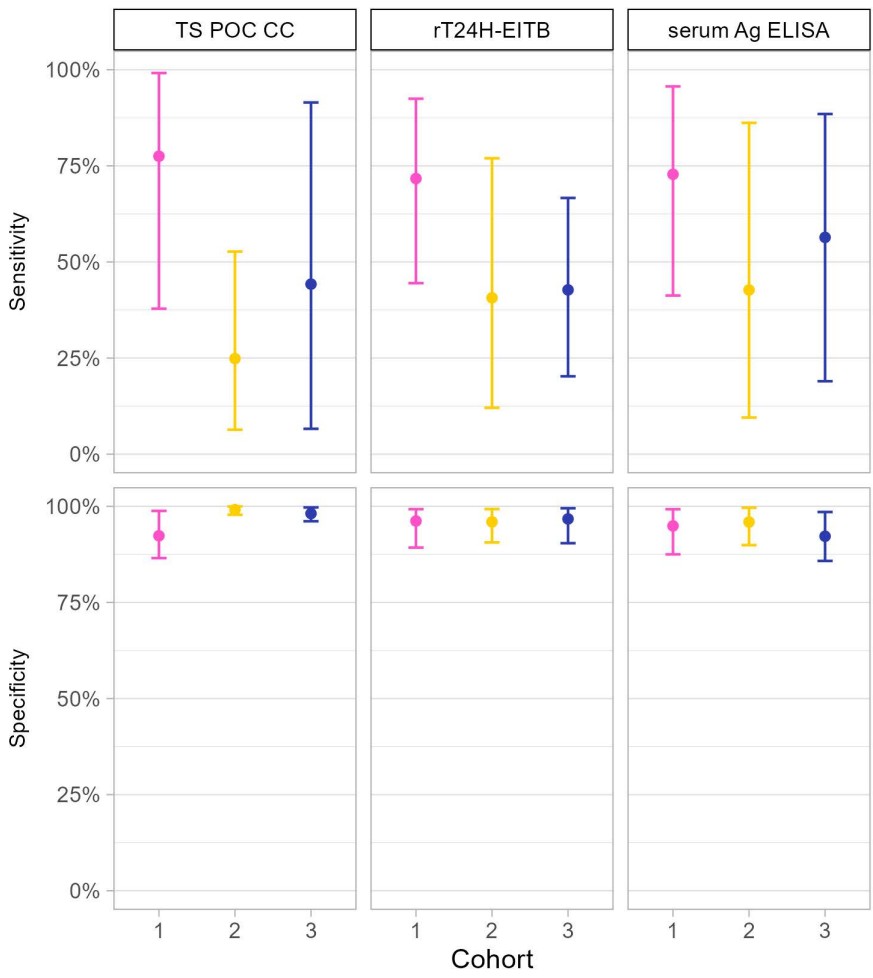

**Fig 5. Sensitivity and specificity of the TS POC CC test strip, rT24H-EITB and serum Ag ELISA to detect cysticercosis in three cohorts recruited in three district hospitals in Tanzania.** Results of the same cohort are indicated with the same color. The error bars indicate the 95% credible intervals around the mean estimates. Cohort 1: participants with specific neurological signs/symptoms compatible with NCC (epilepsy and/or severe progressive headache); cohort 2: participants with complaints compatible with intestinal worm infections; cohort 3: participants with other symptom(s). TS POC CC: *T. solium* point-of-care test for cysticercosis; rT24H-EITB: recombinant enzyme-linked immunoelectrotransfer blot; serum Ag ELISA: serum antigen enzyme-linked immunosorbent assay.

### 3.4.3 Agreement between cysticercosis tests.

Cohen's kappa, positive and negative agreements between each of the different tests are shown in Table 3, for each of the three cohorts. The agreement between the TS POC CC result and the rT24H-EITB was substantial within cohort 1 (κ = 66%), but only fair within the other cohorts (κ = 33-38%). Within each cohort, the agreement of TS POC CC with serum Ag ELISA was lower than the agreement with rT24H-EITB (Table 3). In cohort 1, there was a substantial agreement between the rT24H-EITB and serum Ag ELISA (κ = 74%), whereas the agreement between both tests in the other cohorts was low (κ < 20%). The negative agreements were all above 94%, but the positive agreements were lower, varying between 11% and 76%.

## 4 Discussion

The primary objective of this study was to assess the sensitivity and specificity of the TS POC test under field conditions in people attending district hospitals in Tanzania for the detection of taeniosis and cysticercosis. The study lateral-flow test demonstrated a moderate sensitivity for cysticercosis detection in individuals suspect of (neuro)cysticercosis. In contrast, the sensitivity was largely unsatisfactory for the detection of taeniosis even in clinical suspects. The relatively high specificity of the test may be useful in clinical care.

We used several strategies to minimize bias in this diagnostic accuracy study [13,14]. Despite these efforts, certain factors may have affected the sensitivity and specificity estimates. One potential source of bias for the taeniosis evaluation is the time interval between the initial test and the stool sample collection, which could result in disease progression bias. Although most participants submitted a stool sample within two days of the TS POC test, some waited considerably longer to return a stool sample, raising the possibility of infection acquisition or changes in egg excretion during the interval. In contrast, blood samples were collected promptly by a nurse, thus limiting the risk of disease progression bias. Logistical issues, such as sample shipment and COVID-19-related delays, may have further affected test outcomes. Prolonged storage could have led to false negatives and variations in the stability of antibodies, antigens, and DNA, could have contributed to the observed low agreement between tests. Our two-stage design, sampling all POC test positives and a subset of negatives, aimed to reduce the number of tests. Although weighted adjustments were applied to address partial verification bias, the subset of verified negatives (every 10th patient), may not be fully representative because of the systematic sampling and missing stool or blood samples. Recently, a point-of-care test for the detection of antigens in urine has undergone laboratory validation, showing promise for future field validation [31]. Urine is less invasive than blood sampling and more practical to collect than stool samples, offering potential advantages. The TS POC CC test result was included in the model for taeniosis to account for the two-stage design, but may have influenced results, and complicates the interpretation of the latent class. Due to the lower sensitivity of the TS POC test than the preliminary laboratory tests, we only identified few infections, particularly for taeniosis, which resulted in wide credible intervals for sensitivity. The results should be interpreted with caution as the smaller-than-anticipated analysis population in this study may have affected the reliability and precision of the estimates obtained. Moreover, transforming continuous variables, such as antibody levels or DNA concentrations, into binary outcomes (presence or absence of infection) inevitably leads to a loss of valuable information. This simplification can result in misclassification, particularly near cut-off thresholds. Despite these limitations, this study showed valuable insights in diagnostic tests for *T. solium* infections in hospital settings. Its main strength was evaluating the test in target populations, yielding more realistic estimates than diagnostic case-control studies conducted in laboratory settings.

Rapid diagnostic tests can be particularly useful for identifying *T. solium* taeniosis, as they shorten diagnostic turnaround times and enable quicker treatment, preventing the spread

of infectious eggs. A high sensitivity is essential to avoid missing infected individuals. In our study, the TS POC T test strip had a sensitivity of 40% to 50%, with wide credible intervals, falling short of the 95% sensitivity threshold recommended in Target Product Profiles (TPPs) for monitoring of control interventions, as well as for diagnosis and treatment [8]. Additionally, rES33-EITB, copro Ag ELISA and copro mPCR also showed sensitivities below the 95% TPP threshold, highlighting the need for improved diagnostic methods validated under field conditions. The low disease prevalence complicates sensitivity assessment, as seen in our cohorts where true prevalence was very low and similar to 0.6% [32] and 1.8% [15] in rural communities of Eastern Province of Zambia. Achieving a sensitivity above the 95% TPP threshold for taeniosis tests under field conditions with current tests and low prevalence would thus require very large studies to achieve accurate field validation.

The sensitivity of the TS POC CC test strip for detecting infection with cysticerci varied across the different cohorts. These differences were expected due to varying disease spectra among populations. Sensitivity was low among participants with gastrointestinal symptoms (25% [6 – 53%]) and other symptoms (44% [2 – 95%]), similar to rates observed in asymptomatic rural communities in Zambia (35% [14–63%]) [16]. The sensitivity was higher among participants with neurological signs/symptoms (77% [38-99%]), likely due to higher parasitic loads, resulting in elevated antibody/antigen levels [33], which are more easily detected by a diagnostic test. As such, both rT24H-EITB and serum Ag ELISA also demonstrated higher sensitivities in individuals with neurological signs/symptoms compared to the other cohorts and asymptomatic community members [16]. Consequently, also the agreement between the different cysticercosis tests was higher in individuals with neurological signs/symptoms than in the other cohorts and asymptomatic community members [16]. The almost perfect agreement that has been reported for different NCC tests [34–36] should thus be interpreted cautiously, as it may not be representative for the target population. Differences in test agreement among subgroups stratified by location and stage of the lesions [37] further suggests variability according to disease spectrum. Similarly, the TS POC CC test strip and serological tests showed a sensitivity of 44 to 50% for NCC diagnosis in hospital settings, but above 98% among patients with active lesions [17]. This study shows that agreement between serological tests is only minimal to moderate under field conditions, particularly in individuals not clinically suspect of NCC. In our study, a substantial number of patients were excluded from cohort 1 due to incorrect application of the criteria for severe progressive headache during recruitment. Consequently, the diagnostic estimates may not accurately reflect patients with a presentation of severe progressive headache that differ from the criteria used during screening, potentially limiting the generalizability of our findings to broader patient populations.

There is no gold standard test for diagnosing cysticercosis, and existing tests have been evaluated almost exclusively for neuroimaging-based detection of NCC [33–38], often in (severely diseased) symptomatic patients in diagnostic case-control studies. This leads to an overestimation of the diagnostic performance in laboratory settings [39], making these estimates not representative for the target population. The lack of data on test performance for cysticercosis detection made selecting prior information challenging. Priors were updated based on community-level evaluation data [16], and were chosen to be minimally restrictive, contributing to wide credible intervals. More knowledge on serological test performance under field conditions is urgently needed, not just for NCC.

Compared to the sensitivities, the specificities of all tests were more precisely estimated and generally high, exceeding 90% in all cohorts. When used in a magnetic immunochromatographic test, the rES33 antigen had a specificity of 96% using serum from regions without taeniosis/cysticercosis transmission, with some reactivity in patients infected with *Echinococcus granulosus*, *Ascaris lumbricoides*, *Plasmodium falciparum*, *Trichinella* and *Schistosoma*

*mansoni* [40]. As we evaluated the tests in an endemic region and under field conditions, the specificity estimates that were obtained in our study are likely more realistic than the ones obtained using diagnostic case-control studies. Despite the potential presence of comorbid infections in our study populations, the specificity of the TS POC test seems rather high, and may be useful for epidemiological studies and monitoring of interventions. Nevertheless, when the disease prevalence decreases, a very high specificity is required [41], so the specificity of the tests would have to be evaluated in post-intervention populations to determine if it is fit for purpose.

## 5 Conclusions

The TS POC test demonstrated a moderate sensitivity for infections with cysticerci in individuals with neurological signs/symptoms, but a high specificity, which could be useful for the care of this population. In contrast, the sensitivity was largely unsatisfactory for the detection of *T. solium* taeniosis, even in clinical suspected cases, an infection for which straightforward treatment is available. This study also highlights the lack of knowledge on the performance of the currently used diagnostic tests for *T. solium* taeniosis and cysticercosis under field conditions. Most tests have only been evaluated in laboratory settings using diagnostic case-control studies, resulting in an overestimation of the diagnostic performance of the tests. The variability in test performance across different cohorts underscores the importance of estimating the diagnostic accuracy of a test in the intended target population, and that the sensitivity cannot be extrapolated to populations with a different disease spectrum. Since also the other tests in the current study performed poorly under field conditions, this study showed the overall need to develop more sensitive and specific diagnostic tests to detect human *T. solium* taeniosis and cysticercosis.

## Supporting information

**S1 Appendix. Patient recruitment.** Table 1A Recruitment of cohort 1 participants from the outpatient department and mental health clinic. Table 1B. Symptoms of participants in cohort 1 during recruitment. The results are from screening questionnaires administered by local nurses. S1 Table 1C. Age and gender of patients who were selected for sampling and complete cases for the evaluation of taeniosis. S1 Table 1D. Age and gender of patients who were selected for sampling and complete cases for the evaluation of cysticercosis.
(PDF)

**S2 Appendix. R code for the Bayesian analyses.**
(R)

**S3 Appendix. Output of Bayesian analyses.**
(PDF)

**S1 Checklist. STARD-BLCM-Checklist.**
(DOCX)

## Acknowledgements

We extend our gratitude to all the dedicated hospital staff members and the willing participants for their invaluable contributions to this study. Special recognition is also due to the laboratory technicians, Sandra Vangeenberghe, Maxwell Masuku, Chembensofu Mwelwa, Anke Van Hul, Ana Lucia Fajardo for their diligent analysis of the samples.

Special thanks are due to the members of the SOLID External Advisory Board for their guidance, and Helena Ngowi and Maria V. Johansen for their valuable contributions during

the initial phase of this study. This work is dedicated to the memory of our esteemed colleague and friend, Dr. Benedict Ndawi.

## Author contributions

**Conceptualization:** Chiara Trevisan, Veronika Schmidt-Urbaneja, Kabemba E Mwape, Emmanuel Bottieau, Andrea S Winkler, Pierre Dorny, Pascal Magnussen, Sarah Gabriël, Bernard Ngowi.

**Data curation:** Inge Van Damme, Famke Jansen.

**Formal analysis:** Inge Van Damme, Dries Reynders.

**Funding acquisition:** Kabemba E Mwape, Emmanuel Bottieau, Andrea S Winkler, Sarah Gabriël, Bernard Ngowi.

**Investigation:** Mwemezi Kabululu, Dominik Stelzle, Charles E Makasi.

**Methodology:** Chiara Trevisan, Dominik Stelzle, Kabemba E Mwape, Andrea S Winkler, Pierre Dorny, Sarah Gabriël, Bernard Ngowi.

**Project administration:** Chiara Trevisan, Kabemba E Mwape, Chishimba Mubanga, Karen Schou Møller, Pierre Dorny, Sarah Gabriël, Bernard Ngowi.

**Resources:** Mwemezi Kabululu, Veronika Schmidt-Urbaneja, John Noh, Sukwan Handali, Sarah Gabriël, Bernard Ngowi.

**Supervision:** Chiara Trevisan, Andrea S Winkler, Pierre Dorny, Pascal Magnussen, Sarah Gabriël, Bernard Ngowi.

**Validation:** Inge Van Damme, Chiara Trevisan, Mwemezi Kabululu, Dominik Stelzle, Charles E Makasi, Chishimba Mubanga, Gideon Zulu, Famke Jansen, Sarah Gabriël.

**Visualization:** Inge Van Damme.

**Writing – original draft:** Inge Van Damme, Chiara Trevisan.

**Writing – review & editing:** Mwemezi Kabululu, Dominik Stelzle, Charles E Makasi, Veronika Schmidt-Urbaneja, Kabemba E Mwape, Chishimba Mubanga, Gideon Zulu, Karen Schou Møller, Famke Jansen, Dries Reynders, John Noh, Sukwan Handali, Emmanuel Bottieau, Andrea S Winkler, Pierre Dorny, Pascal Magnussen, Sarah Gabriël, Bernard Ngowi.

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
