## [Decision Letter · Decision Letter 0]

20 Nov 2024

PNTD-D-24-00866Evaluation of a rapid lateral flow assay for the detection of taeniosis and cysticercosis at district hospital level in Tanzania: A prospective multicentre diagnostic accuracy studyPLOS Neglected Tropical Diseases Dear Dr. Van Damme, Thank you for submitting your manuscript to PLOS Neglected Tropical Diseases. After careful consideration, we feel that it has merit but does not fully meet PLOS Neglected Tropical Diseases's publication criteria as it currently stands. Therefore, we invite you to submit a revised version of the manuscript that addresses the points raised during the review process. Please submit your revised manuscript within 60 days Jan 19 2025 11:59PM. If you will need more time than this to complete your revisions, please reply to this message or contact the journal office at plosntds@plos.org. Please include the following items when submitting your revised manuscript: * A rebuttal letter that responds to each point raised by the editor and reviewer(s). You should upload this letter as a separate file labeled 'Response to Reviewers '. This file does not need to include responses to any formatting updates and technical items listed in the 'Journal Requirements' section below. * A marked-up copy of your manuscript that highlights changes made to the original version. You should upload this as a separate file labeled 'Revised Manuscript with Track Changes '. * An unmarked version of your revised paper without tracked changes. You should upload this as a separate file labeled 'Manuscript '. If you would like to make changes to your financial disclosure, competing interests statement, or data availability statement, please make these updates within the submission form at the time of resubmission. Guidelines for resubmitting your figure files are available below the reviewer comments at the end of this letter. We look forward to receiving your revised manuscript. Kind regards, Daniela Fusco, PhDAcademic EditorPLOS Neglected Tropical Diseases Victoria BrookesSection EditorPLOS Neglected Tropical Diseases

Shaden Kamhawi

co-Editor-in-Chief

Paul Brindley

co-Editor-in-Chief

**Journal Requirements:** 1) Please ensure that the CRediT author contributions listed for every co-author are completed accurately and in full. At this stage, the following Authors/Authors require contributions: Inge Van Damme, Chiara Trevisan, Mwemezi Kabululu, Dominik Stelzle, Charles E Makasi, Veronika Schmidt-Urbaneja, Kabemba E Mwape, Chishimba Mubanga, Gideon Zulu, Karen Schou Møller, Famke Jansen, Dries Reynders, John Noh, Sukwan Handali, Emmanuel Bottieau, Andrea S Winkler, Pierre Dorny, Pascal Magnussen, Sarah Gabriël, and Bernard Ngowi. Please ensure that the full contributions of each author are acknowledged in the "Add/Edit/Remove Authors" section of our submission form.The list of CRediT author contributions may be found here: https://journals.plos.org/plosntds/s/authorship#loc-author-contributions 2) Thank you for including an Ethics Statement for your study. Please update it to include:i) A statement that formal consent was obtained (must state whether verbal/written) OR the reason consent was not obtained (e.g. anonymity). NOTE: If child participants, the statement must declare that formal consent was obtained from the parent/guardian.]. 3) Please amend your detailed Financial Disclosure statement. This is published with the article. It must therefore be completed in full sentences and contain the exact wording you wish to be published.1) State the initials, alongside each funding source, of each author to receive each grant. For example: "This work was supported by the National Institutes of Health (####### to AM; ###### to CJ) and the National Science Foundation (###### to AM).".If you did not receive any funding for this study, please simply state: u201cThe authors received no specific funding for this work.u201d 4) For studies involving third-party data, we encourage authors to share any data specific to their analyses that they can legally distribute. PLOS recognizes, however, that authors may be using third-party data they do not have the rights to share. When third-party data cannot be publicly shared, authors must provide all information necessary for interested researchers to apply to gain access to the data. For more information, see https://journals.plos.org/plosntds/s/data-availability#loc-acceptable-data-access-restrictions

For any third-party data that the authors cannot legally distribute, they should include the following information in their Data Availability Statement upon submission: 1) A description of the data set and the third-party source 2) If applicable, verification of permission to use the data set 3) Confirmation of whether the authors received any special privileges in accessing the data that other researchers would not have 4) All necessary contact information others would need to apply to gain access to the data.

**Reviewers' Comments:**Reviewer's Responses to Questions

**Key Review Criteria Required for Acceptance?**

**Methods**

-Are the objectives of the study clearly articulated with a clear testable hypothesis stated?

-Is the study design appropriate to address the stated objectives?

-Is the population clearly described and appropriate for the hypothesis being tested?

-Is the sample size sufficient to ensure adequate power to address the hypothesis being tested?

-Were correct statistical analysis used to support conclusions?

-Are there concerns about ethical or regulatory requirements being met?

Reviewer #1: The study’s objectives—assessing the diagnostic performance of the TS POC test—were clearly articulated, with a testable hypothesis explicitly stated, ensuring a focused and compelling study. Given the absence of a reference test for detecting T. solium taeniosis and cysticercosis, the use of the Bayesian Latent Class Model is a valid and appropriate choice for estimating diagnostic accuracy.

The sample size calculation, as detailed in previous publications, was properly estimated. However, both the prevalence of T. solium and the sensitivity of the test’s performance under laboratory conditions were overestimated, leading to a relatively small number of positive cases for analysis. This issue was appropriately acknowledged by the authors in the discussion section.

Regarding the statistical methodology, I recommend that the authors further clarify and define the targeted the targeted condition (latent class) in the models used for the two tests. For example, it is unclear why the TS POC CC test for cysticercosis is included in the model designed to estimate taeniosis. This inclusion affects the latent class and, consequently, the sensitivity results of the TS POC test under evaluation, which is intended to diagnose taeniosis alone. Similarly, the exclusion of the rES33-EITB test, which targets antibodies, is not sufficiently justified, particularly since the TS POC T test—also targeting antibodies—was included in the model.

The use of directed acyclic graphs, as recommended in the recent paper by Denwood (doi: 10.1371/journal.pntd.0012481), could enhance clarity and assist readers in understanding the model structure.

Reviewer #2: - Are the objectives of the study clearly articulated with a clear testable hypothesis stated? YES

- Is the study design appropriate to address the stated objectives? YES

- Is the population clearly described and appropriate for the hypothesis being tested? YES

- Is the sample size sufficient to ensure adequate power to address the hypothesis being tested? NO

in this study, the analysis populations appear to be very small (much smaller than the sample size estimate) making results very uncertain. When disease prevalence is very low and the number of disease positives is small, as in the present study, the data may not be strong enough to "override" the priors, so the choice of priors can have a significant impact on the results. In low prevalence situations and with 3-4 tests, sample sizes well above 1000 are recommended for reliable estimation. I recommend revisiting if the data allows BLCM.

- Were correct statistical analysis used to support conclusions? In theory yes but given the actual data rather not for reasons as in previous point.

- Are there concerns about ethical or regulatory requirements being met? NO

2 Material and methods

2.1 Study design

l. 146: You introduce the other tests as reference tests though in absence of a gold standard, which would in fact be the reference test, I suggest changing the terminology to avoid confusion. In line with the need for using the BLCM due to the absence of a gold standard, i.e., reference test, none of the other tests can be called reference, and they are equally handled in the model as the TS POC.

l. 148: Same here. I suggest cross-checking with the recommended terminology in the literature. The way it is phrased here it sounds as if the remaining tests form a composite reference standard but this would be in contrast to why you chose a BLCM.

2.4 Sample collection and reference testing

Same comment regarding the use of the terminology of reference test which is usually reserved for the gold standard or composite reference standard.

l.216-218: I suggest moving the details on the test that was not used for this paper to discussion section instead of methods.

2.5.1.1 Handling missing and inconclusive results

l. 247-250: The method for addressing the potential partial verification bias analytically sounds very good. You may consider adding to your limitations that even weighted adjustments might not completely remove bias since the subset selected for further testing is not truly random (due to selecting every 10th patient) though this is only a very small risk in your case. Still, the effectiveness of the correction depends on how representative the subset of verified negatives is.

2.5.3 Sample size and descriptive statistics

The sample sizes are tailored to the diagnostic accuracy estimations though the actual analysis populations for BLCM are far below these numbers. Datasets may be to small to apply BLCM that produce reliable estimates and I recommend critically reflecting on it and at the minimum add a paragraph on the implications for interpretation and limitations in your discussion section.

Reviewer #3: 1. Please can the authors elaborate on the third cohort for testing indicated in lines 157 – 159? It is not apparent my this would be a target population for using the TS POC test.

2. Are the criteria stated in lines 167 – 168 available in the appendix?:

“Different additional criteria were used to be included in the different cohorts.”

And this sentence should be rephrased (with reference to the additional criteria) to something similar to the below to improve clarity:

“Different additional exclusion criteria were used in the different cohorts.”

3. Can the authors state what cut-off was used in line 210: “using a predefined cut-off value”.

4. I am not sure what you mean by “using a pre-determined protocol to determine the cut-off value” on line 214-215.

5. The authors state that on lines 216-218:

“Although also the LLGP-EITB was initially planned to be included in the study [11], the results of this in-house test could not be used due to doubts about the validity of the test results [13]”.

Reviewing reference [13], Mubanga et al. elaborate that “LLGP EITB….they did not satisfy the internal quality control during the laboratory analyses”; can the authors confirm that this was the same issue? The LLGP-EITB is a major diagnostic used for human CC, especially in central/south American settings, so to not include this diagnostic as part of the reference testing needs further justification.

6. Where attempts made to access any neuroimaging based results, since the patient population was derived from a district hospital setting, for the validation of the TS POC CC output (against active infection, rather than just exposure?).

7. Can the authors explain why this statement is necessary on line 219:

“No adverse events were recorded”, is this in relation to the test itself or taking blood for the test?

8. Regarding the statement on lines 254 – 256:

“However, the priors were revised since diagnostic accuracy measures differ according to the target population, and the data did not support the priors in certain models”; With the priors being revised based on the least restrictive set of posteriors generated from the community-based study, it is not clear how these would be relevant for the hospital-based setting (even using the least-restrictive set of posteriors). Can the authors expand on why they think this approach is suitable.

9. It would still be useful to include the model outputs from the combined cohorts (indicated on lines 274 – 276) in the supplementary, even with high Bayesian p values for completeness.

10. Please can the authors explain why a hierarchal model was not used, to account for hierarchical structure of the data (especially that patients were recruited from three different district hospitals, where systematic differences e.g. healthcare infrastructure, patient population, or diagnostic practices may exist?)

**Results**

-Does the analysis presented match the analysis plan?

-Are the results clearly and completely presented?

-Are the figures (Tables, Images) of sufficient quality for clarity?

Reviewer #1: The results are well presented and, with the supplementary materials provided in the appendix, are comprehensive and thorough.

Reviewer #2: 3 Results

I recommend to use the actual analysis populations as basis for the descriptive Table 1 or at least first describe the full cohorts and in the same table also describe the characteristics of the complete case analysis populations. This way you can assess and discuss selection bias that may have been introduced by the strong reduction of the initial cohorts to those individuals that had complete diagnostic test results across all tests under evaluation. I thus also recommend moving Table 1 below the Fig. 1-3 so people already understand the flow of participants.

Table 3: It is recommended to add 95% CI to the estimates.

General comments to BLCM models:

You may reconsider using BLCM for your data in general. The underlying analysis populations and test positivity rates are very low causing a large uncertainty regarding all estimates. This is an indication that the model estimates are instable and e.g., senstitivities almost across the whole range of values from 0-100 are also plausible results given your data. Please critically revisit if the data is suitable to run BLCM. The choices of priors are very restrictive and seem to dominate the model outputs regarding specificities almost fully ignoring the underlying data. Were the tests all considered to be conditionally independent or dependent? Please specify your assumptions on the dependence structure in the methods section and clarify how it was handled in the analysis.

Reviewer #3: 1. In terms of those individuals not included from the pool of “Potentially eligible patients” in cohort 1, due to “incorrect headache criteria used”, the constitutes a reasonably large proportion (138/742) – can the authors expand on this limitation, especially in the discussion.

2. It would be useful in Tables 2 and 4 to include the number of results under the first 4 columns, especially where there are negative/positive combinations.

**Conclusions**

-Are the conclusions supported by the data presented?

-Are the limitations of analysis clearly described?

-Do the authors discuss how these data can be helpful to advance our understanding of the topic under study?

-Is public health relevance addressed?

Reviewer #1: The authors clearly acknowledge the study's methodological limitations in the discussion section and carefully consider them when formulating their conclusions.

Reviewer #2: - Are the conclusions supported by the data presented? BLCM estimates are very uncertain which makes evidence on diagnostic accuracy of the tests based on the empirical data rather weak.

- Are the limitations of analysis clearly described? NO, limitations of the methods applied and the scarce data and low prevalence are not sufficiently adressed.

- Do the authors discuss how these data can be helpful to advance our understanding of the topic under study? YES

- Is public health relevance addressed? YES

Reviewer #3: This section is well addressed - but see specific comments relating to the Discussion and Conclusion sections under "Summary and General Comments"

**Editorial and Data Presentation Modifications?**

Reviewer #1: Minor Revision

I suggest that the authors further clarify the target condition underlying the models to improve the interpretation of the diagnostic performance estimates.

Reviewer #2: (No Response)

Reviewer #3: The data and outputs are generally well presented, I include some minor recommendations in the "Results" section.

**Summary and General Comments**

Reviewer #1: (No Response)

Reviewer #2: Abstract

Methods

The number of diagnostic tests should be part of the abstract already. How many coprological and serological tests were performed in parallel per patient? This is informative to later interpret results from the BLCM. Also already mention that you assessed your BLCM varying the prior specifications.

Results

Are the populations in whom the TS POC was conducted also complete for the other tests? Please specify N of analysis populations for BLCM as that appears to be the core of the manuscript. The section only focusses on TS POC results though other tests performance was assessed too. Will be good to mention the others performance too. Add info on what is represented in square brackets, 95% Credible Intervals?

Discussion

I suggest replacing ‘suboptimal sensitivity’ by ‘low sensitivity’ since 5 out of 6 estimates are <50%. I suggest rephrasing the last sentence where you recommend ‘for the development and validation of better diagnostic tests’ by sth like ‘to enhance and validate these tests for better performance in practical, real-world settings’.

1 Background

l. 123-129: I suggest moving most of what you have not done to the discussion and rather focus on the objectives of this paper as described from l. 130 in the background section. Please mention which other tests are being assessed along with the TS POC which is your main interest.

Reviewer #3: This is an important study, with a suitable methodology to address the problem. I have outlined a few points of clarification required for the methods before I would suggest this manuscript is ready for acceptance/minor revisions. Other comments include for specific sections (generally minor comments):

Introduction

Major

1. It is not clear to me how this study differs from reference 11 which is also a study of the diagnostic performance in district hospitals (level 2?) in Tanzania as reported in the statement (lines 125 – 126):

“To evaluate its real-world performance in resource-poor, highly endemic areas in sub-Saharan Africa, the performance characteristics of the TS POC test were evaluated in two different settings: …….., and in district hospitals in Tanzania [11]”

Please can the authors make this clarify and make the distinction clearer in the introduction.

Minor

1. On lines 100-101, it would be useful to include the prevalence ranges from refs 1 and 2: “In Tanzania, the prevalence of T. solium taeniosis and cysticercosis is high [1,2]”

2. Please provide references for examples of different infectious diseases to support this statement on lines 112 – 113:

“Rapid diagnostic tests (RDTs) have revolutionized the field of diagnostics by

providing fast and easy-to-use tools for the detection of diseases in resource-limited settings”

3. Please provide a reference to support the following statement on lines 121-123:

“The test showed a promising sensitivity and specificity to detect T. solium taeniosis and NCC during its preliminary evaluation under laboratory conditions”

Methods

Minor

11. A brief sentence or two to explain the overall trial how the diagnostic study relates to the overall trial, after lines 137 – 138, would be useful.

12. Please replace “neither” with “either” in line 147 to remove the double-negative in the sentence.

13. Please provide a reference for the statement on lines 189 – 194 or if references 17 and 18 are relevant, please use the references here in addition:

“During initial assessments conducted within laboratory settings using known positive and negative control sera, the TS POC test demonstrated encouraging results. The TS POC T test strip exhibited a sensitivity of 82% and a specificity of 99% in detecting taeniosis. In the case of NCC, the TS POC CC test strip had a sensitivity of 88%, and 93% for infection with multiple cysticerci, with a corresponding specificity of 99%.”

14. Please change “nor” to “or” on line 207 to avoid the double negative.

15. Please change “is” to “are” on line 265

Results

Minor

1. In reference to 3.3.2 and 3.4.2 in estimation of diagnostic performance, these sections are based on the BLCM modelling? Please can the authors state this to link the results more clearly to the methods section.

Discussion

Major

1. Can the authors explain why they think the sensitivity of the Copro-Ag ELISA is higher across the cohorts compared to the Copro mPCR molecular test?

2. The authors could expand on the usefulness of the tests in ruling out disease (with a negative test) given the high specificity and the applicability related to this.

3. While the samples were shipped and processed in Belgium, which the authors suggest could raise the issue of increased false negatives, can the authors indicate why it was not possible to process samples in-country, presumably due to limited infrastructure, and therefore what attempts are being made to strengthen capacity in-country to support processing of samples that require lab processing?

4. How does the sensitivity of the TS POC CC test in cohort 1 in this study compare to individuals from community-based studies (which will be a mixture of symptomatic and asymptomatic populations) in relation to lines 517 – 525?

5. The authors should compare their findings with this POC to the POC test based on urine samples indicated by Toribio et al. 2023 and Toribio et al. 2024 (see below). Although this test is based on antigen detection, it would be important to compare the performance and potential utility of both tests:

Toribio L, Handali S, Marin Y, Perez E, Castillo Y, Bustos JA, O'Neal SE, Garcia HH. A Rapid Point-of-Care Assay for Cysticercosis Antigen Detection in Urine Samples. Am J Trop Med Hyg. 2023 Feb 6;108(3):578-580. doi: 10.4269/ajtmh.22-0598.

Toribio LM, Vásquez A, Castillo Y, Salas SM, Perez E, Bustos JA, O'Neal SE, Garcia HH. Concordance between a New Rapid Point-Of-Care Assay and Standard ELISA in the Detection of Cysticercosis Antigens in Urine. Am J Trop Med Hyg. 2024 Aug 13;111(4):823-825. doi: 10.4269/ajtmh.24-0171.

6. The text in the conclusion section (lines 564 – 67):

“Additionally, transforming a continuous variable, such as antibody levels or DNA

concentrations, into a binary outcome (presence or absence of infection) leads to a loss of valuable information, which can result in misclassification and diminish the overall accuracy of a diagnostic test.”

Would be better placed and expanded on in the discussion section.

PLOS authors have the option to publish the peer review history of their article (what does this mean? ). If published, this will include your full peer review and any attached files.

**Do you want your identity to be public for this peer review?** For information about this choice, including consent withdrawal, please see our Privacy Policy .

Reviewer #1: No

Reviewer #2: No

Reviewer #3: No

**Figure resubmission:**While revising your submission, please upload your figure files to the Preflight Analysis and Conversion Engine (PACE) digital diagnostic tool, https://pacev2.apexcovantage.com/. PACE helps ensure that figures meet PLOS requirements. To use PACE, you must first register as a user. Registration is free. Then, login and navigate to the UPLOAD tab, where you will find detailed instructions on how to use the tool. If you encounter any issues or have any questions when using PACE, please email PLOS at figures@plos.org. Please note that Supporting Information files do not need this step. If there are other versions of figure files still present in your submission file inventory at resubmission, please replace them with the PACE-processed versions. **Reproducibility:**To enhance the reproducibility of your results, we recommend that authors of applicable studies deposit laboratory protocols in protocols.io, where a protocol can be assigned its own identifier (DOI) such that it can be cited independently in the future. Additionally, PLOS ONE offers an option to publish peer-reviewed clinical study protocols. Read more information on sharing protocols at https://plos.org/protocols?utm_medium=editorial-email&utm_source=authorletters&utm_campaign=protocols

---

## [Decision Letter · Decision Letter 1]

26 Feb 2025

PNTD-D-24-00866R1Evaluation of a rapid lateral flow assay for the detection of taeniosis and cysticercosis at district hospital level in Tanzania: A prospective multicentre diagnostic accuracy studyPLOS Neglected Tropical Diseases Dear Dr. Van Damme, Thank you for submitting your manuscript to PLOS Neglected Tropical Diseases. After careful consideration, we feel that it has merit but does not fully meet PLOS Neglected Tropical Diseases's publication criteria as it currently stands. Therefore, we invite you to submit a revised version of the manuscript that addresses the points raised during the review process. Please submit your revised manuscript within 30 days Mar 28 2025 11:59PM. If you will need more time than this to complete your revisions, please reply to this message or contact the journal office at plosntds@plos.org. Please include the following items when submitting your revised manuscript: * A rebuttal letter that responds to each point raised by the editor and reviewer(s). You should upload this letter as a separate file labeled 'Response to Reviewers '. This file does not need to include responses to any formatting updates and technical items listed in the 'Journal Requirements' section below. * A marked-up copy of your manuscript that highlights changes made to the original version. You should upload this as a separate file labeled 'Revised Manuscript with Track Changes '. * An unmarked version of your revised paper without tracked changes. You should upload this as a separate file labeled 'Manuscript '. If you would like to make changes to your financial disclosure, competing interests statement, or data availability statement, please make these updates within the submission form at the time of resubmission. Guidelines for resubmitting your figure files are available below the reviewer comments at the end of this letter. We look forward to receiving your revised manuscript. Kind regards, Daniela Fusco, PhDAcademic EditorPLOS Neglected Tropical Diseases

Victoria Brookes

Section Editor

Shaden Kamhawi

co-Editor-in-Chief

Paul Brindley

co-Editor-in-Chief

**Journal Requirements:**

Please ensure that the funders and grant numbers match between the Financial Disclosure field and the Funding Information tab in your submission form. Note that the funders must be provided in the same order in both places as well.

**Reviewers' comments:** Reviewer's Responses to Questions

**Key Review Criteria Required for Acceptance?**

**Methods** :

-Are the objectives of the study clearly articulated with a clear testable hypothesis stated?

-Is the study design appropriate to address the stated objectives?

-Is the population clearly described and appropriate for the hypothesis being tested?

-Is the sample size sufficient to ensure adequate power to address the hypothesis being tested?

-Were correct statistical analysis used to support conclusions?

-Are there concerns about ethical or regulatory requirements being met?

Reviewer #1: The authors added appendices to address previous recommendations. They also expanded the limitations in the conclusions to acknowledge the latent class definition problem.

Reviewer #2: First, I would like to thank the authors for their thorough revision of the manuscript. The responses demonstrate careful consideration of the comments and substantial effort has been made to address each point raised. The expanded discussion and clarifications have notably improved the manuscript.

While I understand the word limit constraints, I have concerns about the prioritisation of information in your manuscript:

1. Abstract and Methods:

While you mention space constraints, I believe some critical information about the current study design should take precedence over contextual information about the larger project. Specifically: (1) The number of diagnostic tests used per participant; (2) the analysis population sizes for BLCM. These are essential details for readers to understand and interpret your results properly.

2. Background Section:

While I appreciate that this study is part of a larger project, I maintain that the current manuscript should prioritise details relevant to this specific analysis. I suggest: (1) Moving broader project context to a single paragraph with appropriate references.

These suggestions aim to ensure that readers have access to all essential information about the current study's design and analysis, while still adhering to word limits. Context about the larger project, while valuable, should not come at the expense of key methodological details.

In addition, I have few remaining concerns in the methods and results section:

Regarding Table 3 and Confidence Intervals:

Your response suggests that calculating CIs is "not straightforward" due to the weighting in the two-stage design. However, this doesn't necessarily mean it's impossible. There are established methods for calculating confidence intervals for weighted estimates, including bootstrap methods that could account for the complex sampling design. Could you either: (1) Implement one of these methods to provide CIs, or (2) More explicitly explain why these standard approaches for weighted data would not be applicable in your specific case?

Regarding BLCM and Prior Influence:

Your response about the prior influence raises several concerns: (1) Your last sentence mentions that "the analysis allowed the sensitivities and specificities to differ conditional on the other test results," but this needs more elaboration. This appears to address the question about conditional dependence, but it's not clear how this was implemented in the model (dependence in positives, dependence in negatives, dependence in both?). (2) Not all details of the BLCM and modelling diagnostics are provided. Hence, it is not possible to assess whether the right approach was adopted and to reproduce the results. I suggest that the code is included in the SI.

Reviewer #3: Questions/points raised have been well addressed. Thank you.

**Results** :

-Does the analysis presented match the analysis plan?

-Are the results clearly and completely presented?

-Are the figures (Tables, Images) of sufficient quality for clarity?

Reviewer #1: (No Response)

Reviewer #2: (No Response)

Reviewer #3: Questions/points raised have been well addressed. Thank you.

**Conclusions** :

-Are the conclusions supported by the data presented?

-Are the limitations of analysis clearly described?

-Do the authors discuss how these data can be helpful to advance our understanding of the topic under study?

-Is public health relevance addressed?

Reviewer #1: (No Response)

Reviewer #2: (No Response)

Reviewer #3: Questions/points raised have been well addressed. Thank you.

**Editorial and Data Presentation Modifications?**

Reviewer #1: (No Response)

Reviewer #2: (No Response)

Reviewer #3: Questions/points raised have been well addressed. Thank you.

**Summary and General Comments** :

Reviewer #1: (No Response)

Reviewer #2: (No Response)

Reviewer #3: Questions/points raised have been well addressed. Thank you.

PLOS authors have the option to publish the peer review history of their article (what does this mean? ). If published, this will include your full peer review and any attached files.

**Do you want your identity to be public for this peer review?** For information about this choice, including consent withdrawal, please see our Privacy Policy .

Reviewer #1: No

Reviewer #2: No

Reviewer #3: **Yes: ** Matthew A. Dixon

---

## [Editor Report · Decision Letter 2]

3 Mar 2025

Dear Dr. Van Damme,

We are pleased to inform you that your manuscript 'Evaluation of a rapid lateral flow assay for the detection of taeniosis and cysticercosis at district hospital level in Tanzania: A prospective multicentre diagnostic accuracy study' has been provisionally accepted for publication in PLOS Neglected Tropical Diseases.

Best regards,

Daniela Fusco, PhD

Academic Editor

Victoria Brookes

Section Editor

Shaden Kamhawi

co-Editor-in-Chief

Paul Brindley

co-Editor-in-Chief

---

## [Editor Report · Acceptance letter]

Dear Dr. Van Damme,

We are delighted to inform you that your manuscript, "Evaluation of a rapid lateral flow assay for the detection of taeniosis and cysticercosis at district hospital level in Tanzania: A prospective multicentre diagnostic accuracy study," has been formally accepted for publication in PLOS Neglected Tropical Diseases.

Best regards,

Shaden Kamhawi

co-Editor-in-Chief

Paul Brindley

co-Editor-in-Chief
